# Hypoxia Is Associated with Increased Immune Infiltrates and Both Anti-Tumour and Immune Suppressive Signalling in Muscle-Invasive Bladder Cancer

**DOI:** 10.3390/ijms24108956

**Published:** 2023-05-18

**Authors:** Vicky Smith, Dave Lee, Mark Reardon, Rekaya Shabbir, Sudhakar Sahoo, Peter Hoskin, Ananya Choudhury, Timothy Illidge, Catharine M. L. West

**Affiliations:** 1Division of Cancer Sciences, University of Manchester, Manchester M13 9PL, UK; 2Computational Biology Support, CRUK Manchester Institute, Alderley Park SK10 4TG, UK; 3The Christie Hospital NHS Foundation Trust, Manchester M20 4BX, UK; 4Mount Vernon Cancer Centre, Northwood HA6 2RN, UK; 5Manchester Academic Health Science Centre, Manchester M13 9NQ, UK

**Keywords:** hypoxia, HIF, immune TME, TIME, bladder cancer, MIBC, ChIP-seq, immunotherapy, ICIs

## Abstract

Hypoxia and a suppressive tumour microenvironment (TME) are both independent negative prognostic factors for muscle-invasive bladder cancer (MIBC) that contribute to treatment resistance. Hypoxia has been shown to induce an immune suppressive TME by recruiting myeloid cells that inhibit anti-tumour T cell responses. Recent transcriptomic analyses show hypoxia increases suppressive and anti-tumour immune signalling and infiltrates in bladder cancer. This study sought to investigate the relationship between hypoxia-inducible factor (HIF)-1 and -2, hypoxia, and immune signalling and infiltrates in MIBC. ChIP-seq was performed to identify HIF1α, HIF2α, and HIF1β binding in the genome of the MIBC cell line T24 cultured in 1% and 0.1% oxygen for 24 h. Microarray data from four MIBC cell lines (T24, J82, UMUC3, and HT1376) cultured under 1%, 0.2%, and 0.1% oxygen for 24 h were used. Differences in the immune contexture between high- and low-hypoxia tumours were investigated using in silico analyses of two bladder cancer cohorts (BCON and TCGA) filtered to only include MIBC cases. GO and GSEA were used with the R packages “limma” and “fgsea”. Immune deconvolution was performed using ImSig and TIMER algorithms. RStudio was used for all analyses. Under hypoxia, HIF1α and HIF2α bound to ~11.5–13.5% and ~4.5–7.5% of immune-related genes, respectively (1–0.1% O_2_). HIF1α and HIF2α both bound to genes associated with T cell activation and differentiation signalling pathways. HIF1α and HIF2α had distinct roles in immune-related signalling. HIF1 was associated with interferon production specifically, whilst HIF2 was associated with generic cytokine signalling as well as humoral and toll-like receptor immune responses. Neutrophil and myeloid cell signalling was enriched under hypoxia, alongside hallmark pathways associated with Tregs and macrophages. High-hypoxia MIBC tumours had increased expression of both suppressive and anti-tumour immune gene signatures and were associated with increased immune infiltrates. Overall, hypoxia is associated with increased inflammation for both suppressive and anti-tumour-related immune signalling and immune infiltrates, as seen in vitro and in situ using MIBC patient tumours.

## 1. Introduction

In the UK, the standard-of-care treatment for muscle-invasive bladder cancer (MIBC) patients is either radical cystectomy or radiotherapy with a radiosensitiser, and neoadjuvant chemotherapy if the patient is fit enough [1]. In addition to direct cancer-cell killing, radiotherapy effectiveness depends on eliciting an anti-tumour immune response driven by dendritic cells (DCs) and T cells. However, radiotherapy can also induce a pro-tumour inflammatory response by the proportional increase in regulatory T cells (Tregs) alongside the release of cytokines and chemokines that recruit myeloid cell populations such as neutrophils, macrophages, and myeloid-derived suppressor cells (MDSCs) [2,3]. Recruited Tregs and myeloid cells suppress anti-tumour T cell responses and contribute to disease progression and recurrence [4]. Therefore, an existing suppressive tumour microenvironment (TME) potentiates the pro-tumour capabilities of radiotherapy-induced immune responses and is a poor prognostic factor for radiotherapy outcomes [3]. Hypoxia is also a poor prognostic factor that contributes to radiotherapy resistance, disease progression, and recurrence in many solid tumours, including bladder cancer [5,6,7].

Cellular responses to hypoxia are mostly regulated by hypoxia-inducible factor (HIF) transcription factors, which are heterodimers consisting of alpha and beta subunits. There are three different HIFs, driven by different HIF-α isoforms binding to the HIF1β subunit, of which HIF1 and HIF2 are the best studied [8]. Hypoxia has been linked to driving a suppressive immune TME by altering the phenotypes and activities of different immune cells [9]. Hypoxia inhibits antigen uptake of DCs and alters their cytokine and chemokine expression, which reduces T cell activity and increases neutrophil recruitment to create a suppressive immune TME [10]. Hypoxia has also been shown to inhibit neutrophil apoptosis to prolong their normal survival time and promotes MDSC inhibition of T cell proliferation and their differentiation into suppressive tumour-associated macrophages (TAMs) [11,12]. Moreover, HIF can drive the inflammatory potential of neutrophils and TAMs [13], and the latter are found in the highest densities in hypoxic regions and tend to have a T-cell-inhibiting suppressive M2 phenotype [14,15,16]. Hypoxia and the associated adenosine accumulation can also inhibit CD8^+^ T cell proliferation and infiltration into hypoxic areas and induce CD8^+^ T cell apoptosis [17,18,19,20]. Little has been reported specifically for human bladder cancer, aside from a study showing macrophage infiltration positively correlated with HIF1α expression, angiogenesis, and a poor prognosis [21].

Conversely, HIF1α has been shown to play an essential role in inducing and maintaining CD8^+^ T cell effector state functions to enhance CD8^+^ T-cell-mediated tumour killing [22,23]. Recently, a meta-analysis investigated the relationship between *HIF1A* gene expression and the immune TME in ten TCGA cohorts. The authors showed that in bladder cancer there was a positive correlation between *HIF1A* expression and both immune suppressive (PD-L1, Tregs, MDSCs, and M2 macrophages) and anti-tumour immune (CD8^+^ T cells, NK cells, M1 macrophages, and IFN response) gene signatures [24].

Given the lack of study in bladder cancer, the overall aim of this study was to investigate the relationship between HIF, hypoxia, and immune-related signalling in MIBC. The first objective was to investigate hypoxia-associated and HIF-specific regulation of immune-related genes and signalling pathways in MIBC using in vitro approaches. The second objective was to investigate differences in immune signalling and infiltrates between high- and low-hypoxia MIBC using in silico approaches.

## 2. Results

### 2.1. ChIP-Seq Identified HIF Binding Sites with High Specificity and Low Background

ChIP sequencing was performed to identify genome-wide binding sites for HIF1α, HIF2α, and HIF1β in the T24 MIBC cell line cultured under 1% and 0.1% oxygen. Appendix A shows the heatmaps of the input, HIF1α, HIF2α, and HIF1β signal intensities. The figure shows a high signal for each sample at the transcriptional start sites (TSS) and high specificity compared to input background signal intensity. Appendix A also shows an enrichment of mapped reads around TSS for each sample over the input control, further illustrating the specificity of the ChIP samples. Peaks were filtered according to four different parameters: all significant peaks; protein coding peaks; near-TSS peaks; and both protein coding and near-TSS peaks (hereby termed stringent). Appendix A shows that different numbers of peaks were identified when comparing oxygen concentrations (0.1% vs. 1%) and samples (HIF1α vs. HIF2α vs. HIF1β). According to the highest stringency filtering level, there were more HIF-bound genes at 0.1% vs. 1% oxygen, and approx. 3-fold more genes bound by HIF1α than HIF2α (Appendix A).

### 2.2. HIF1 and HIF2 Are Associated with Distinct Biological Processes

The large number of peaks (Appendix A) made analysis at the individual gene level difficult. Therefore, over-representation analysis was performed using the genes identified by the most stringent filtering level to look at gene sets found more frequently than expected by chance. As expected, the top 20 gene sets enriched for HIF1β included processes associated with metabolism and oxygen level (Figure 1A,B). HIF1α and HIF2α associated with distinct biological processes, which differed depending on the severity of hypoxia (Figure 1). The top 20 enriched gene sets for HIF2α included myeloid cell differentiation (1% oxygen) and TGF-β signalling (0.1% oxygen).

### 2.3. HIF1 and HIF2 Are Associated with Unique Immune-Related Processes

To identify which HIF-bound genes are immune-related, the EBI QuickGO resource was used to cross-reference ChIP-seq identified genes with those annotated as “immune response”. The proportion of HIF-bound genes that were immune-related was higher for HIF1α than HIF2α and increased as the oxygen concentration decreased from ~4.8–11.8% in 1% oxygen for HIF2α and HIF1α, respectively, to ~7.5–13.4% in 0.1% hypoxia (Table 1). Appendix A shows that a number of these immune-related genes were unique to either HIF1α or HIF2α. Only HIF2α was enriched at an enhancer region of the PD-L1 gene (*CD274*) under 1% and 0.1% hypoxia, which was visualised using the University of California Santa Cruz (UCSC) genome browser resource (Appendix A). The majority of peaks identified at 1% oxygen were also present at 0.1% oxygen (Appendix A). Over-representation analysis was performed on the subunit unique genes to identify enriched immune-related gene sets. There were differences in the top 20 gene sets for the unique immune-related genes bound to each subunit. HIF1α was associated with signalling related to adaptive immune responses such as interferon-associated signalling (Figure 2A,B); and HIF2α with signalling related to innate immune responses such as humoral and toll-like receptor signalling (Figure 2C,D). The top 20 enriched immune-related pathways for both HIFs included T cell activation/differentiation (Figure 2).

### 2.4. Hypoxia Associates with Myeloid, Neutrophil, and CD4^+^ T Cell Signalling Processes

Microarray transcriptomics was used to investigate differentially expressed genes (DEGs) under hypoxia (0.1%, 0.2%, and 1% O_2_) compared to normoxia (21% O_2_) in four MIBC cell lines (T24, J82, UMUC3, and HT1376). Gene Ontology (GO) over-representation analysis was used to investigate DEGs (*p* < 0.1) that were significantly (*p* < 0.05) enriched for biological processes under the GO search term “immun” for any of the cell lines under each oxygen concentration. Biological processes associated with myeloid and neutrophil signalling were enriched in cells cultured in all three low oxygen concentrations (0.1% O_2_ shown in Table 2; 1% and 0.2% shown in Appendix A, respectively). Gene set enrichment analysis (GSEA) using the hallmark pathways geneset showed that under hypoxia Hallmark_TNFα_signalling_via_NFkB (1%, 0.2%, 0.1% O_2_) and Hallmark_IL2_STAT5_signalling (1%, 0.2% O_2_) were in the top ten significantly enriched pathways alongside Hallmark_hypoxia and Hallmark_glycolysis (1%, 0.2%, 0.1% O_2_) and Hallmark_epithelial_to_mesenchymal_transition signalling (0.2%, 0.1% O_2_; Figure 3).

### 2.5. Hypoxia Associates with an Inflamed TME in MIBC Patient Tumours

To assess how hypoxia affects immune signalling in human tumours in situ, BCON and TCGA-BLCA MIBC gene expression datasets were used to correlate hypoxia scores with the expression of immune signalling pathways. The bladder cancer 24-gene hypoxia gene signature was correlated with the scores of various immune-related gene signatures. Heatmaps show that there is an increased expression of the immune-related gene signatures in high-hypoxia tumours (hypoxia scores greater than the median), and low expression in low-hypoxia samples in the BCON (Figure 4A) and TCGA cohorts (Figure 4B). Boxplots show that MIBC with high versus low hypoxia has significantly increased expression of the immune-related signatures, apart from mast cell signalling and NK cell signalling in the BCON cohort (Appendix A). ImSig and TIMER immune cell deconvolution algorithms assessed the presence of immune cell infiltrates for low vs. high hypoxia tumours in the BCON and TCGA-BLCA datasets. As shown in Figure 4, high-hypoxia tumours had significantly more T cells and neutrophils, as shown by both algorithms. ImSig further shows that high-hypoxia tumours had significantly more monocytes and NK cells, whilst TIMER showed significantly more myeloid dendritic cells. Macrophages were significantly increased in high-hypoxia tumours when analysed by ImSig but were not significantly different when using TIMER for the BCON cohort (Figure 4C,E). Macrophages were significantly increased in high-hypoxia tumours, as seen by both algorithms in the TCGA cohort (Figure 4D,F). There were differences in B cell infiltrate levels for the two algorithms; with TIMER showing a decrease in hypoxic tumours (not significant in TCGA) and ImSig showing a significant increase (Figure 4).

## 3. Discussion

There are several novel findings from this study regarding the role of HIF and hypoxia in immune-related processes in MIBC. First, we show that HIF1 and HIF2 bind uniquely to some immune-related genes in the T24 MIBC cell line, which was associated with distinct immune-related processes, as demonstrated by the finding that only HIF2α bound to an enhancer region of the PD-L1 gene. Second, we find that high-hypoxia tumours have an increased presence of immune infiltrates compared to low hypoxia. Our work also consolidates the findings showing that hypoxia upregulates signalling related to both anti-tumour and immune-suppressive pathways in multiple cancers including bladder cancer [24], which we now show in MIBC cohorts specifically.

In a similar manner to this study, Symthies et al. performed a HIF ChIP-seq experiment on kidney and liver cancer cell lines cultured in 0.5% and 3% oxygen [25]. Smythies et al. also found that oxygen concentration did not alter HIF binding locations but increased the strength of binding, as we have demonstrated here. Our finding of a higher proportion of binding sites for HIF1α compared to HIF2α is also consistent with the findings of others in renal, breast, and liver human cell lines [25,26,27].

Smythies et al. found that HIF1 and HIF2 heterodimers bound to distinct regions of the genome without competing and this was conserved across four human cancer cell lines (HKC-8 and RCC4, renal; HepG2, liver; and MCF-7, breast) [25]. The work here showed that HIF1 and HIF2 associate first with common processes of oxygen consumption and sugar glycolysis and then with distinct biological processes. Smythies et al. also showed that wherever HIF1β bound it was with an HIF-α isoform, in concordance with the published literature [25,28]. The HIF1β sample obtained in this study is of worse quality than the HIFα isoforms, as shown in Appendix A. The lower quality decreased the number of significant genes bound by HIF1β, compared to HIF1α and HIF2α that made it through the stringent filtering. However, the results for the subunit unique immune binding sites showed that most of the HIF1β binding sites overlap with either of the HIF-α isoforms, confirming the binding of the subunits to form heterodimers as expected.

The results presented in this report show for the first time in MIBC that HIF1 and HIF2 bind to some unique immune-related genes. These results show that ~10% of all immune-related genes are bound by HIF in the T24 MIBC cell line. Although it is known that HIF has a role in directly regulating many immune-related genes, a comprehensive list of HIF-regulated immune genes has not been generated previously, so no comment can be made on whether this proportion of immune-related gene binding is expected. The published literature tends to focus on the immune-suppressive effects of HIF and its binding of specific genes that contribute towards immune evasion mechanisms due to their important effects on tumour progression and resistance to treatments [3]. Previous studies have shown that HIF1α binds to the PD-L1 promoter, and Noman et al. further showed that HIF2α does not, in prostate, breast, and melanoma cell lines [29,30]. In this study, there were binding peaks for both HIF-α subunits when using the most lenient filtering parameter but only HIF2α is retained when using stringent filtering. Under both 1% and 0.1% oxygen, HIF2α binding was enriched for an enhancer region of the PD-L1 gene (*CD274)*. Studies investigating the mechanisms governing PD-L1 expression at a genomic level give rise to discrepancies and have rarely included bladder cancer [31]. As discussed in our previous research, there are potential differences in the interaction between HIF and PD-L1 across different tissue types, so this discrepancy is likely to be cell line/cancer-type dependent [32]. A recent study by Bruns et al. showed that *HIF1A* induced *CD274* expression in TCGA lung cancer but neither breast nor melanoma cancers which further indicates the potential for tissue-specific HIF regulation of PD-L1 [33]. A study analysing the role of HIF1α and -2α in inducing PD-L1 expression suggested that in kidney cancer, HIF2α is the main regulator of PD-L1 expression and not HIF1α [34]. Additional studies are needed to explore the interaction between HIF and PD-L1 in more MIBC cell lines and to further elucidate the molecular mechanisms of PD-L1 expression overall.

The results presented here show that the immune-related processes most enriched by HIF binding in vitro are those associated with immune-stimulatory pathways. Interferon is a class of cytokines that has a key role in the induction of anti-tumour immune responses [35]. Enrichment of immune-related pathways revealed different immune-related activities between HIF1 and HIF2. Unique HIF1α immune-related processes were enriched for positive regulation of various interferon signalling pathways and T cell activation and differentiation. Unique HIF2α immune-related processes were enriched for humoral responses, generic cytokine and chemokine regulation (some negative), complement activation, some innate immune responses such as toll-like receptor and lipopolysaccharide-sensing signalling, and also T cell activation and differentiation. Non-unique over-represented immune-related pathways for HIF2α specifically also included associations with immune suppressive roles such as myeloid cell differentiation and TGF-β signalling. The enrichment of these immune pathways implies a broader role for HIF2α, whilst HIF1α was enriched for pathways involved in the stimulation of anti-tumour immune responses. These results are in agreement with the published literature showing the role of both HIFs but mostly HIF1α in the activation and effector functions of T cells [22,23].

Expanding on the HIF-specific results, immune-related signalling in a panel of MIBC cells under hypoxic conditions was enriched for myeloid and neutrophil signalling as seen by gene ontology analysis. Whilst still being fully elucidated, TNFα is known to have a role in tumour-promoting immune signalling via the induction of NF-kB [36]. TNFα via NF-kB has been shown to inhibit anti-tumour immune responses of leukocytes and to contribute to tumour cell proliferation, migration, and metastasis [37]. TNFα regulates macrophage activation and function and can induce pro-inflammatory cytokine signalling [38]. IL-2 STAT5 signalling has a role in the differentiation of CD4^+^ cells, which is mostly well-characterised for its role in maintaining Treg differentiation [39,40]. As shown by GSEA analysis using hallmark pathways, both TNFα via NF-kB and IL2 STAT5 signalling was significantly enriched under hypoxia, along with EMT and hypoxia-related signalling. These results indicate the potential difference between HIF-dependent and hypoxia-associated effects on immune-related signalling by tumour cells. As considerable cross-talk occurs between immune cells present in the TME, it is important to expand from in vitro analysis to consider relationships between the immune TME and hypoxia in the context of patient tumours.

Different immune gene signatures were used to associate immune signalling with hypoxia using transcriptomic data for MIBC from the BCON and TCGA cohorts. A 24-gene bladder cancer hypoxia gene signature assigned tumours as high- or low-hypoxia [41]. Heatmaps showed that tumours assigned as high hypoxia were associated with higher expression of both immune-suppressive (checkpoint, TGFβ -ECM, M2 TAM, exhausted CD8, macrophage, and neutrophil) and anti-tumour (M1 TAM, cytotoxic, DC, NK cell, and T cell) gene signatures. Boxplots confirmed the statistical significance of the high versus low hypoxia increases in immune-related signature expression. Hypoxia-associated increases in tumour inflammation are supported by a study performed by Chen et al. analysing ten different cancer types including bladder cancer [24].

To investigate if tumour hypoxia affects the presence of immune infiltrates, two different immune cell deconvolution algorithms were used, ImSig and TIMER [42,43]. There was a high level of concordance between the two cohorts and algorithms, with the exception of B cells where hypoxia was associated with increases using ImSig and decreases using TIMER. All of the other immune infiltrates (monocytes, macrophages, DCs, neutrophils, NK cells, and T cells) increased significantly in MIBC assigned as high hypoxia versus low hypoxia. These results are further supported by three recently published bladder cancer hypoxia-associated prognostic gene signatures. All three studies showed that tumours assigned as hypoxic had increased infiltration of various immune cells and enrichment of immune-related signalling [44,45,46].

The work presented here is limited by the use of only one MIBC cell line and would benefit from further ChIP-seq experiments on different MIBC cell lines. The lack of analysis at the protein level is also a limitation of this study.

In conclusion, HIF1 and HIF2 associate with distinct immune-related signalling in MIBC but this is likely to be tissue-type dependent and requires further elucidation. The current literature indicates that hypoxia has an immune-suppressive role in a TME. The work here shows that hypoxia increases both suppressive and anti-tumour-related immune signalling and highlights the need to consider the balance between the two when analysing hypoxia-driven immune signalling. Further work is needed to investigate the mechanisms and differences between HIF-dependent and HIF-independent hypoxia-related immune signalling in MIBC.

## 4. Materials and Methods

### 4.1. Cohorts

BCON was a prospective multicentre phase III clinical trial that recruited patients in the UK from 2000 to 2006 (registered as CRUK/01/003), of which the trial protocol and results are described in detail elsewhere [47]. Transcriptomic data (*n* = 152) were generated previously as detailed elsewhere [41] and the updated long-term clinical outcomes were used throughout [48]. RNAseq data from the TCGA bladder cancer cohort (*n* = 405) was obtained using the R packages “TCGAUtils” and “curatedTCGAData”. TCGA (*n* = 401) and BCON (*n* = 141) datasets were filtered to include only tumours stage 2 and above, i.e., MIBC.

### 4.2. ChIP-Seq Data Generation

T24 bladder cancer cells were cultured for 24 h in both 0.1% and 1% O_2_. The protein–DNA interactions were cross-linked using ChIP cross-link gold (Diagenode, Denville, NJ, USA) and 1% formaldehyde before lysing the cells and shearing the chromatin into 200–300 bp fragments using a Biorupter Pico (Diagenode). Antibodies against HIF1α, HIF2α, and HIF1β, and Dynabeads Protein G were used for immunoprecipitation (Appendix A). The fragments were de-cross-linked and the DNA was eluted using the phenol–chloroform method. DNA with no immunoprecipitation was processed and sequenced in parallel as the input control. A qPCR was used to validate the ChIP experiment before the samples were sequenced and mapped by the CRUK Manchester Institute core facilities. Sequencing reads for all samples underwent quality control assessment and adapter removal with FASTQC [49] and Trim Galore [50] software, respectively. Trimmed fastq files were mapped against the hg19 reference assembly using bowtie2 with 1 allowed mismatch in seed alignment (-N set to 1). Resulting SAM files were converted into BAM format with samtools. Peaks were called with MACS2 software and subsequent annotation of identified peaks was performed with Homer (v4.10) where peak-to-gene annotations used the genes nearest to the transcriptional start site.

### 4.3. Microarray Data Generation

Microarray data were generated for a panel of MIBC cell lines (T24, J82, UMUC3, and HT1376) under various oxygen concentrations (21%, 1%, 0.2%, and 0.1%). Cells were cultured for 24 h in each condition and RNA was extracted using RNeasy Plus Mini Kit (Qiagen). Gene expression arrays were generated using Clariom S pico HT human assay (Thermo Fisher, Waltham, MA, USA) by Yourgene Health and batch-corrected using ComBat function from the R package “sva” to produce log_2_ summarised gene level expression.

### 4.4. Data Analysis

R and RStudio were used throughout, alongside the package “tidyverse”. All ChIP-seq data analysis was performed using the most stringent filtering parameter (peaks close to transcriptional start site and protein coding). Over-representation analysis was performed using the “clusterProfiler” package to generate the top 20 significant (adjusted *p*-value < 0.05) gene ontology biological processes and graphically represented using “enrichplot”.

Gene signatures from the published literature were used and hypoxia scores were assigned using the Yang et al. bladder cancer hypoxia gene signature. Median scores across this panel of genes formed the basis for stratifying cohorts into low and high hypoxia groups [41].

The R package “limma” was used to obtain differentially expressed genes (DEGs; *p* < 0.1) across any of the cell lines in each oxygen concentration compared to normoxia. The function “goana” was used with the DEGs to investigate gene ontologies annotated using the search term “immun” that were significantly (*p* < 0.05) enriched under hypoxia. The R package “fgsea” was used to perform the GSEA with hallmark pathways from “msigdb” and the DEGs to investigate which hallmark pathways were significantly (*p* < 0.05) enriched under hypoxia.

ImSig was applied using the R package “ImSig” [42] and TIMER deconvolution was performed using the website http://timer.cistrome.org/ (accessed on 16 February 2022) with BLCA as the cancer type [43].

## Figures and Tables

**Figure 1 ijms-24-08956-f001:**
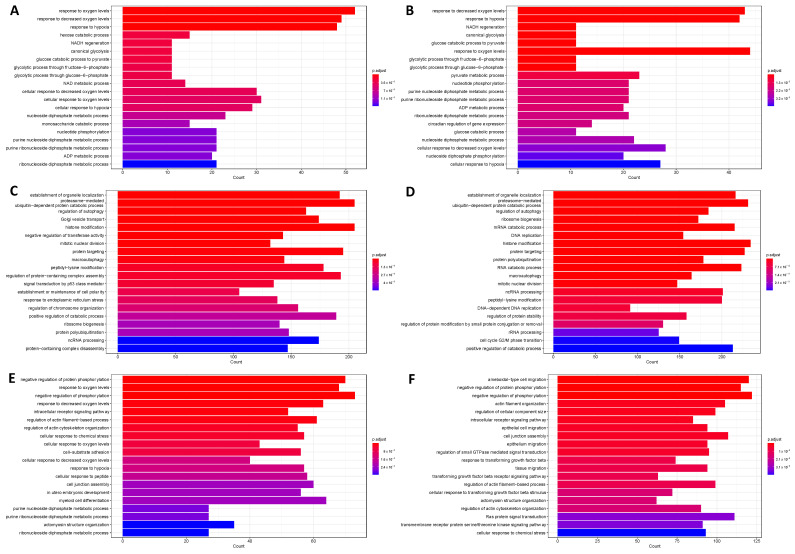
HIF1β targeted pathways enriched in (**A**) 1% and (**B**) 0.1% oxygen; HIF1α targeted pathways enriched in (**C**) 1% and (**D**) 0.1% oxygen; HIF2α targeted pathways enriched in T24 cells cultured in (**E**) 1% and (**F**) 0.1% oxygen. Enriched gene ontology (GO) biological processes terms were identified with R package “clusterProfiler”. Each term was ordered according to statistical significance (BH) and the top 20 results were visualised as bar plots. *x*-axis refers to the number of HIF1β bound genes from the dataset that were mapped onto that given GO term.

**Figure 2 ijms-24-08956-f002:**
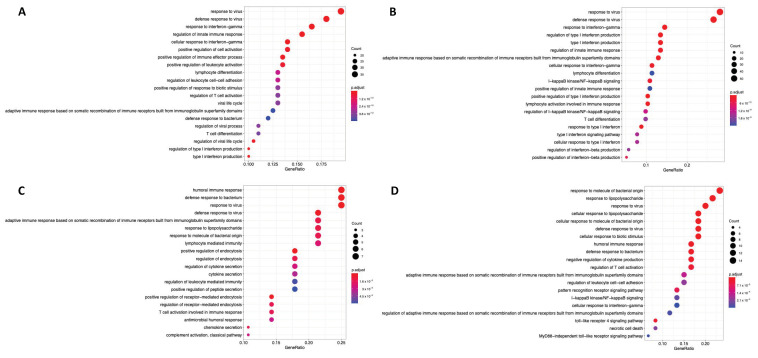
Over-representation analysis for HIF1α unique immune genes in (**A**) 1% and (**B**) 0.1% oxygen, and HIF2α unique immune genes in (**C**) 1% and (**D**) 0.1% oxygen. Enriched gene ontology (GO) biological processes terms were identified with R package “clusterProfiler”. The top 20 terms were plotted and ordered according to count. Count is the number of genes in this dataset that mapped onto the given GO term. *x*-axis is the gene ratio, which is the count divided by the total number of genes annotated to the given GO term, presented as a ratio.

**Figure 3 ijms-24-08956-f003:**
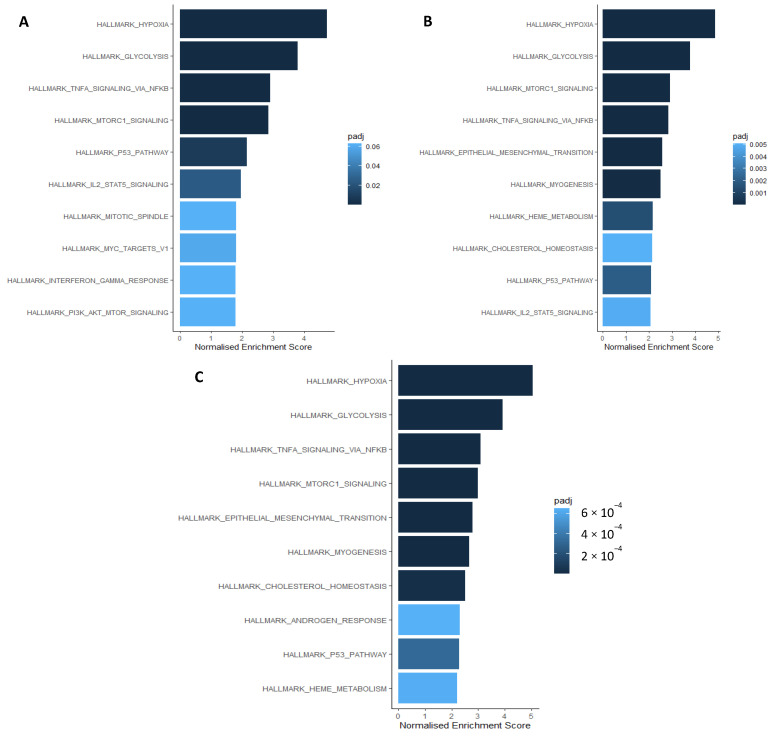
Gene set enrichment analysis showing the hallmark pathways significantly enriched under (**A**) 1%, (**B**) 0.2%, (**C**) 0.1% hypoxia ordered according to normalised enrichment score. R package “fgsea” was used for the analysis and significance was defined as *p*-value of <0.05, with adjusted *p*-values shown in the figure legend using the colour key.

**Figure 4 ijms-24-08956-f004:**
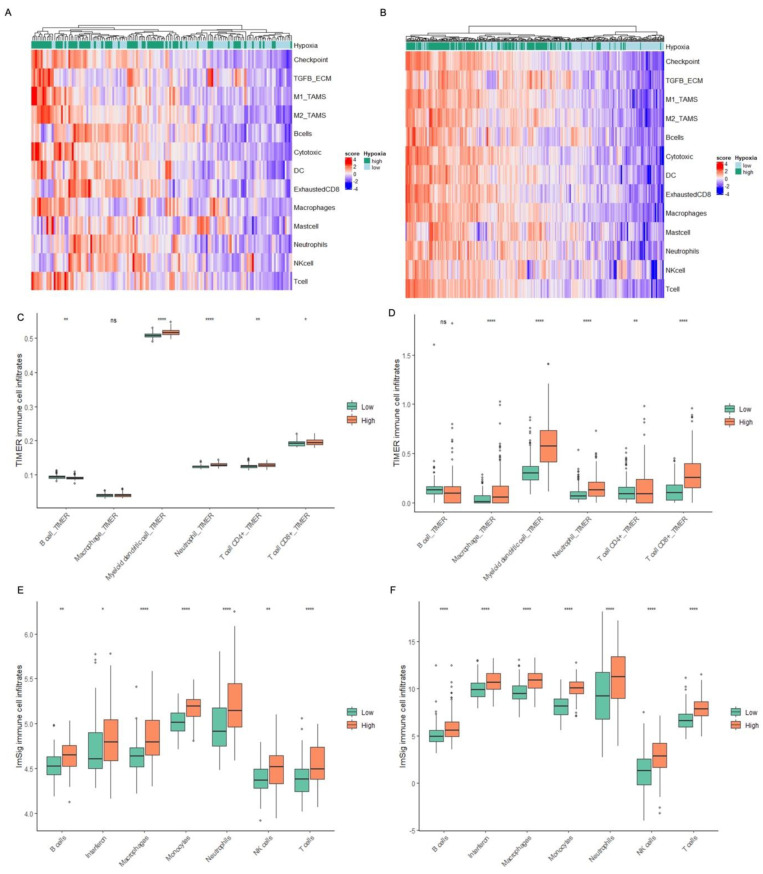
Heatmaps showing the clustering of immune-related signature scores in relation to hypoxia status, high or low, in (**A**) BCON and (**B**) TCGA cohorts. Boxplots showing the fraction of immune cell population according to hypoxia status as deconvoluted by (**C**) TIMER, (**E**) ImSig for the BCON cohort; and (**D**) TIMER, (**F**) ImSig for the TCGA cohort. The R package “ComplexHeatmap” was used to generate the heatmap. Hypoxia status was stratified by the median hypoxia score of the cohort. Statistics are *p*-values from *t*-tests represented as: ns = not significant, * *p* < 0.05, ** *p* < 0.01, **** *p* < 0.0001.

**Table 1 ijms-24-08956-t001:** Percentage of immune-related genes bound to each HIF subunit.

Oxygen Concentration	HIF Subunit	Percent of Immune Genes Bound
1%	HIF1α	11.79
	HIF1β	2.20
	HIF2α	4.77
0.1%	HIF1α	13.43
	HIF1β	1.76
	HIF2α	7.54

Values are the percentages of genes annotated as immune-related by EBI Quick GO (*n* = 2494) for genes identified as bound by each HIF subunit according to the stringent filtering level.

**Table 2 ijms-24-08956-t002:** GO terms filtered by the search term “immun” significantly enriched under 0.1% hypoxia.

Term	ID	Ont	*n*	DE	P.DE
mitigation of host immune response by virus	GO:0030683	BP	2	2	0.04
positive regulation of tolerance induction dependent upon immune response	GO:0002654	BP	2	2	0.04
positive regulation of immune response to tumour cell	GO:0002839	BP	13	6	0.03
positive regulation of myeloid leukocyte cytokine production	GO:0061081	BP	19	8	0.02
Neutrophil-mediated immunity	GO:0002446	BP	501	127	<0.001
neutrophil activation involved in immune response	GO:0002283	BP	490	121	0.003
myeloid-leukocyte-mediated immunity	GO:0002444	BP	555	136	0.003
myeloid cell activation involved in immune response	GO:0002275	BP	549	127	0.02

Ont is the gene ontology term, BP = biological process. *n* = number of genes in the GO term. DE = number of differentially expressed genes from dataset present in the GO term. P.DE = *p*-value for over-representation of the GO term in the set.

## Data Availability

Data are available upon request from Vicky Smith. BCON data are available upon reasonable request from Peter Hoskin.

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
