# Peer review of "Hypoxia Is Associated with Increased Immune Infiltrates and Both Anti-Tumour and Immune Suppressive Signalling in Muscle-Invasive Bladder Cancer"

_ijms, 2023, doi:10.3390/ijms24108956_

Round 1
Reviewer 1 Report
Smith et al. examines the correlation between hypoxia and immune regulatory genes/immune cell infiltration in muscle invasive bladder cancer. While in concept the manuscript is of interesting and worthwhile topic, the current work falls short in that the findings are mostly repetitive of what has been seen in other tumor types, and there are no mechanism studies. The manuscript is essentially based on 1-2 single experiments, and there is little or no causative experiments to follow up on the CHIPseq data. The novelty of the manuscript is low, and there should be some attempt by the authors to enhance the mechanistic insights between hypoxia, HIF localization to immune regulatory genes, and actual immune infiltration.
Author Response
Thank you for taking the time to review this manuscript and for your comments.
We agree that the manuscript is an interesting and worthwhile topic and we think that it offers several novel findings that are worthy of publication, despite the lack of mechanistic studies.
Firstly, the interaction between HIF transcription factors and immune genes in MIBC has not been shown before and is useful knowledge for others studying the interactions between HIF/hypoxia and MIBC. The ChIPseq dataset will be made available to any and all reasonable requests for it.
Secondly, studies investigating the relationship between hypoxia and MIBC are few and far between and we think this also lends novelty to the paper that could further research in this field. We agree that further mechanistic studies would greatly benefit this research topic, but that these would fall beyond the scope of this predominantly bioinformatics investigative manuscript.
Thirdly, there is interest in the effects of hypoxia on PD-L1 expression due to the implications for immune checkpoint and hypoxia-modification therapies. This paper shows interesting insights into how HIF specifically interacts with PD-L1 at the genomic levels, of which nothing has been shown in bladder cancer previously and literature is extremely limited for other cancer and tissue types.
Therefore, we believe that this manuscript provides interesting and useful knowledge that fills gaps in the relevant research field, making it a useful addition to current literature. We hope that it provides a resourceful base from which other more mechanistic studies and investigations can arise.
Reviewer 2 Report
This predominantly bioinformatics investigation of hypoxia and HIF signatures in bladder cancer supports previous studies of these and other authors which have described associations between hypoxia/HIF signalling and immune function in a variety of cancer types. Without back-up confirmation of results with protein analysis or experimental studies, the findings remain speculative. However, similar reports across multiple publications and in several cancer types tend to consolidate assertions of a link between hypoxia and immune signalling that can facilitate tumour growth via multiple pathways, including subsets that may be tissue-specific. Overall, the study is well-conceived and presented, figures and tables are all presented and annotated appropriately, and referencing is suitable for this type of bioinformatics investigation. I recommend the manuscript for publication in its present form as it is almost free of typographical and grammatical errors, and I feel that inclusion of the types of experiments that will be required to support these preliminary findings would result in the manuscript becoming too long (and therefore better communicated in a separate publication).
1. Supplementary Table 2: Please state distance from the TSS that was used to define ‘near TSS’.
Author Response
Thank you for taking the time to review this manuscript and thank you for your comments .
We agree that further mechanistic and investigative work would be beneficial to the topic, but that it falls beyond the scope of this predominantly bioinformatics investigation. We believe that publication of this manuscript will enable these further studies and investigations into this little-researched area to further fill the gap in current literature around this topic and specifically in MIBC.
I have now added in the definition of ‘near TSS’ in the relevant table, where it is defined as in Homer software by within -1000bp and +100bp of an annotated TSS.
Reviewer 3 Report
Dear Authors,
The manuscript submitted is referring, if I have correctly understood, to previously collected data and samples, first published in 2010. If it is correct, It should be very interesting to add the follow up of the patients in order to better understand the role of hypoxya. Furthermore, since the patients involved were of both gender, additional pathologies, if present, should be added. For example, how much prostatitis, or prostate cancer, can be involved.
Minor concerns:
- 4.3. subheading should be the 4.1.
- HIF1 and/or HIF2 should be written always in the same way and not HIF-1 and/or HIF-2.
Author Response
Thank you for taking the time to review this manuscript and thank you for your comments.
The data does have follow up data for the patients. We did consider looking into the follow up data for the patients for this manuscript, but it made it too lengthy and we felt like it fell outside of the aim of this study. However, we took this study forwards to develop a hypoxia-driven immune gene signature, which we show has prognostic significance in multiple MIBC cohorts, predicts benefit of hypoxia-modifying therapy (from the BCON dataset) and associates with increased immune-associated signalling. It is currently being assessed on ICI cohorts (ImVigor dataset and in collaboration with AZ) for prediction of benefit and will be written up as a manuscript in due course. We think that evaluating the patient follow up is more suited to the gene signature study and due to this it falls outside the scope of the results presented in this manuscript here.
In response to your minor concerns I have made the following changes:
- Inserted ‘4.3 Cohorts’ at the start of the methods section so it is now section 4.1 instead and 4.1 and 4.2 are shifted down to 4.2 and 4.3, respectively.
- I have edited all written instances of HIF-1 or HIF-2, including HIF-1α, HIF-2α, and HIF-1β to read without the hyphen e.g. HIF1α.